# Thermodynamic Analysis of Partitioned Combined Cycle using Simple Gases

**Aqeel Ahmad Taimoor [1,\*], Muhammad Ehtisham Siddiqui [2]** and **Salem S. Abdel Aziz [2,3]**

1   Department of materials and chemical engineering, Ghulam Ishaq Khan Institute of Engineering Sciences and Technology, Swabi 23460, Pakistan

2   Mechanical Engineering Department, King Abdulaziz University, Jeddah 21589, Saudi Arabia; ehtisham.siddiqui@gmail.com (M.E.S.); salemanos@kau.edu.sa (S.S.A.A.)

3   Mechanical Power Engineering Department, Faculty of Engineering, Zagazig University, Sharkia 44519, Egypt

\*   Correspondence: taimooruet@gmail.com

**Abstract:** In combined cycle gas turbines, most of the energy loss is usually due to the high temperature of the exhaust gases. Different heat recuperation methods are used. In this study, a novel direct method for heat recovery is investigated. Confidence in the results is established by accounting for all the losses and simulation errors while comparing with the conventional cycle. Aspen HYSYS and MATLAB are the simulation tools used. The General Electric (GE) 9HA.02 combined cycle is taken as a base case. Five gases, air, argon, hydrogen, nitrogen, and carbon dioxide, are studied with the proposed modification. The efficiency maximization function is updated and the pressure and temperature ratios of individual Brayton and Rankine cycles are discussed. The combustor/heat exchanger is modified and simulated according to the known principles of heat and momentum transfer. The whole simulation algorithm is provided. Equation of state (EOS-PR) is used to calculate the properties at every discretized step (for $H_2$, critical properties are modified/HYSYS inbuilt feature). Different gases are analyzed according to their property profiles over the whole cycle. The effect of fluid properties on efficiency is discussed as a guideline for any tailored fluid.

**Keywords:** combined cycle; heat recuperation; optimization

## 1. Introduction

In stationary combined cycle gas turbines (CCGT), the Rankine cycle is used commercially as a bottoming cycle. The high exhaust temperature in the Brayton cycle instigates immense energy loss. This heat is partially recovered in the Rankine cycle. The efficiency of the modern Brayton cycle is higher than the Rankine cycle (fluid exergy loss is less in the Brayton cycle than the Rankine cycle as the latter requires latent heat at low pressures). Therefore, empirically high heat reclamation to work in the Brayton cycle enhances the overall efficiency. Due to some associated Rankine cycle limitations, several variants have been proposed to enhance the overall efficiency of the gas turbine (GT). One effective approach is the regeneration cycle (RG). Direct or indirect methods can be used to recover heat before rejection of flue gases to the bottoming cycle (if one is available). Direct methods employ a simple heat exchange concept and recover heat, where temperature difference and component efficiencies are favorable. Thermodynamically, it is most pragmatic to exchange heat after compression and before the combustion chamber in a cycle operating above atmospheric pressure.

Indirect methods usually involve steam generation/injection or endothermal chemical reactions. Steam injection is usually employed to reduce oxides of nitrogen (NOx), although it has been reported to augment the power output [1–11] but it results in latent heat and entropy loss associated with

working fluid. Any efficiency enhancement by this technique is only reported for the simple Brayton cycle. Moreover, combustors should be modified (larger design is required) before steam injection utilization [12–14]. Furthermore, a closed cycle (like the Rankine cycle) eclipses it because of water loss, as downstream water recovery from flue gases is not a convenient option. Although, condenser addition is suggested, but the effects of acid gases condensation are not accounted [15].

Any chemical reaction intrinsically involves the entropy loss and lowers the advantages of heat recuperation (compared to direct heat recovery). Nakagaki et al. used exergy rates to explain this aspect [16]. Mostly, recuperation involves reforming reactions [17–19] or gasification reactions [20–24]. Ahmed et al. studied the benefits of integrating both reforming and gasification along with $CO_2$ capture [25]. The incorporation of fuel cells is also reported to enhance GT efficiency [26–29]. Carapellucci concluded, based on his characteristic plane technique, that simple indirect heat recovery techniques are not adequate. Thus, indirect methods must be used in combination to harness any benefit. Additionally, he provides different ranges for heat recuperation usage for retrofitting the Brayton cycle to become comparable with CCGT [30].

Several previous investigations signify direct heat recuperation as a commercially viable approach, provided an efficient heat exchanger is present. Direct heat exchange with a combustion chamber has not been reported but preheating of the combustion fuel and air with exhaust has been extensively studied for GT. Excessive pressure drop in the heat exchanger is specifically important [31]. Practically, transferring heat involves extensive pressure drops. Therefore, earlier studies showed that direct heat recuperation requires pressure ratios optimization [32]. Pressure ratios also manifest an effect on the compressor discharge temperature. Furthermore, intercooling of the compressor enhances the heat regeneration gain (mainly because of high temperature gradients in the downstream heat exchanger) [33,34].

Xiao et al. reviewed recuperators for micro GTs and reported that an effective exchanger should have high effectiveness (>90%) and less pressure drop (<3%) [35]. Sayyaadi et al. [36,37], and Avval et al. [38] applied the direct energy recuperation on a 60 MW Siemen's gas turbine (Brayton cycle) and 106 MW using a tubular exchanger. They optimized the cost and efficiency. They proposed to retrofit the turbine by using empirical optimization techniques, although the exchanger pressure drop is calculated and inculcated in the model but waste heat recovery with a bottoming cycle is not evaluated. Moreover, the gas mixture was assumed as an ideal. New gas turbines offer high temperatures and pressures. Taking working fluid at these conditions as an ideal gas can seriously affect the conclusions. Salpingidou et al. discussed the recuperation at different levels of turbine stages for an aero Gas Turbine (GT) [39]. Tubular exchanger design as a recuperator was also discussed [40]. The heat recuperation concept was also applied on Combined Cycle Gas Turbine (CCGT) [41–43].

The double acting Stirling cycle has also been integrated, using a heat exchanger, with GT and reported to increase the efficiency for low wattage power generators [44]. Poullikkas proposed that the combustion chamber of a GT can be used as a hot reservoir for the Stirling cycle [45]. Durante et al. modeled (using the Number of Transfer Units (NTU)-effectiveness method) a high temperature (>1123 K) ceramic heat exchanger for an externally fired gas turbine. In externally fired gas turbines, heat is transferred from an external combustion reaction to the working fluid in the Brayton cycle [46]. The same principle is applied for GTs, operating on nuclear energy. Printed circuit heat exchangers (PCHE) are also reported to exchange heat with working fluids [47]. PCHE allow gas–gas heat exchange with an effectiveness of more than 90% [48–51]. However, the drawback lies in the huge pressure drop. Thus, PCHE are mostly suitable to the heavily pressurized systems like supercritical carbon dioxide or to power cycles with working fluid that offers less exergy loss due to pressure drop [52,53].

The idea of heat recuperation is due to the Brayton cycle waste heat, which results because of:

1.   Non ideality of the working fluids, resulting in entropy loss (high exhaust temperature);
2.   Adiabatic loss owing to moving machinery;

3.  A pressure ratio that cannot be optimized to recover all the heat present in the working fluid in the Brayton cycle;

4.  High excess air to combustors because of the metallurgical temperature limit.

The exergy analysis conducted on CCGT and on its variants optimized the loss because of second law. Therefore, the underlined principles of the first two points are well defined and waste heat cannot be avoided beyond a certain limit. In our previous publication [54], it was shown that pressure ratios are chosen as a function of turbine inlet temperature. There is an optimum pressure ratio for each combustion temperature in CCGT.

Theoretically, a higher temperature in the combustion chamber always favors the thermal efficiency. The maximum temperature that can be achieved is adiabatic flame temperature, but the material constraints limit this temperature approach. High excess air, used for lowering the temperature, can be curtailed by using exhaust gas recirculation, but compressor exergy loss because of high exhaust temperature does not allow it beyond certain threshold. Furthermore, any additional fluid offering less thermodynamic loss than excess air cannot be used, as its recovery from the exhaust mixture poses a challenge. In this study, a novel approach is discussed to limit excess air and recover energy before it manifests in the form of waste heat. The objective is to maximize heat recovery from the flue gases in CCGT. This partially eliminates the need for recuperation. The advantage of the presented approach is that high temperature differences for the heat exchanger are ensured, which results in its size reduction. Moreover, the recuperation concepts are applied on relatively small wattage turbines. This article, on the other hand, focuses on a 755 MW CCGT.

ASPEN Tech. process simulators are reported to imitate the CCGT (with added variations) process conditions within the acceptable range [55,56]. The same was used to conduct this study. The gas turbine model was validated using the data from the General Electric 9HA.02 turbine. The turbine had an installed capacity of 755 MW [54].

## 2. Conceptual Design

The simple CCGT is shown in Figure 1. Air and fuel are compressed and mixed in the combustion chamber. Axial compressors are used to avoid any abrupt change in the fluid direction, to avoid excessive pressure drops. The burner is designed to ensure proper mixing of fuel and air. Excess air is used to maintain the adiabatic flame temperature. The cooling effect of the air is also desirable, to avoid excessive NOx production. Exhaust from the turbine is at a high temperature. Rejecting these gases to the atmosphere without heat recuperation results in a huge energy penalty. Therefore, in stationary GTs, the Rankine cycle is typically used as a bottoming cycle to utilize this waste heat. Usually, a flame less heat recovery steam generator (HRSG) is used to exchange heat between the vaporizing water and the exhaust gases. The exhaust gases cannot be cooled below a certain limit to avoid the acid dew point. The dew point is because of the water in the gases produced due to combustion reaction and humidity. This water, if condensed, dissolves oxygen and acid gases resulting from the burning of the foul compounds (present in the fuel), causing corrosion of the waste heat boiler cold spots [57]. Furthermore, an extensively foul fuel requires a scrubber to meet stringent environment limits [58]. The two cycles, the Brayton cycle and the Rankine cycle, are commercially termed as CCGT.

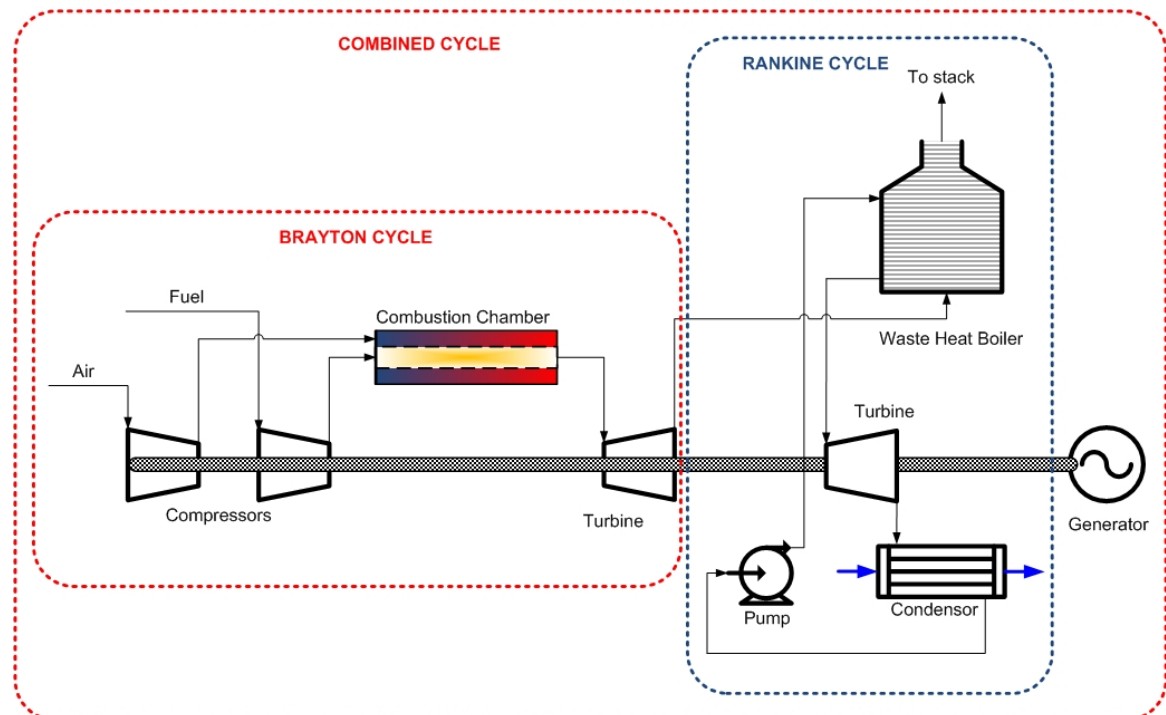

**Figure 1.** Pictorial representation of a simple combined cycle, including an open Brayton cycle and a bottoming closed Rankine cycle.

The excess air in the Brayton cycle, although it serves to cool, curtail NOx, and enhance fuel mixing, also introduces a thermodynamic loss. In the proposed cycle, as shown in Figure 2, it is suggested to keep air close to the stoichiometric amounts. The direct outcome of excess air curtailment is the temperature rise in combustors. The combustor chambers are modified into heat exchangers. The excess heat is exchanged between the CCGT and partitioned combined cycle gas turbine (PCGT). The combustion chamber/heat exchanger is designed to avoid any mixing of the fluids between the two cycles. The fluids exchange heat without altering the gases' direction (parallel flow) to avoid excessive pressure drops. The exchanged heat is then used to run a PCGT. As the main CCGT is divided, each requires an optimized individual Rankine cycle. This is due to the different conditions of the two Brayton cycles. Thus, consequently the Rankine cycle is also partitioned. PCGT working fluid can be exploited with the following two options:

1.  Choosing a fluid with better exergy than the combustion mixture;
2.  As the PCGT working fluid does not have any water of combustion, its dew point will be much higher. This will practically make it possible to lower the exhaust temperature of the flue gases, resulting in better cycle efficiencies.

The PCGT working fluid can be tailored and optimized. In order to understand/estimate the behavior of fluids in PCGT we start with simple gases with a range of heat capacities, densities, and conductivities to relate to energy provision, volumetric, and heat transfer properties, respectively. The rest of the article discusses different thermodynamic pros and cons of this proposed PCGT scheme compared to CCGT. The focus is to design the equipment within a realistic range for high-wattage gas turbines.

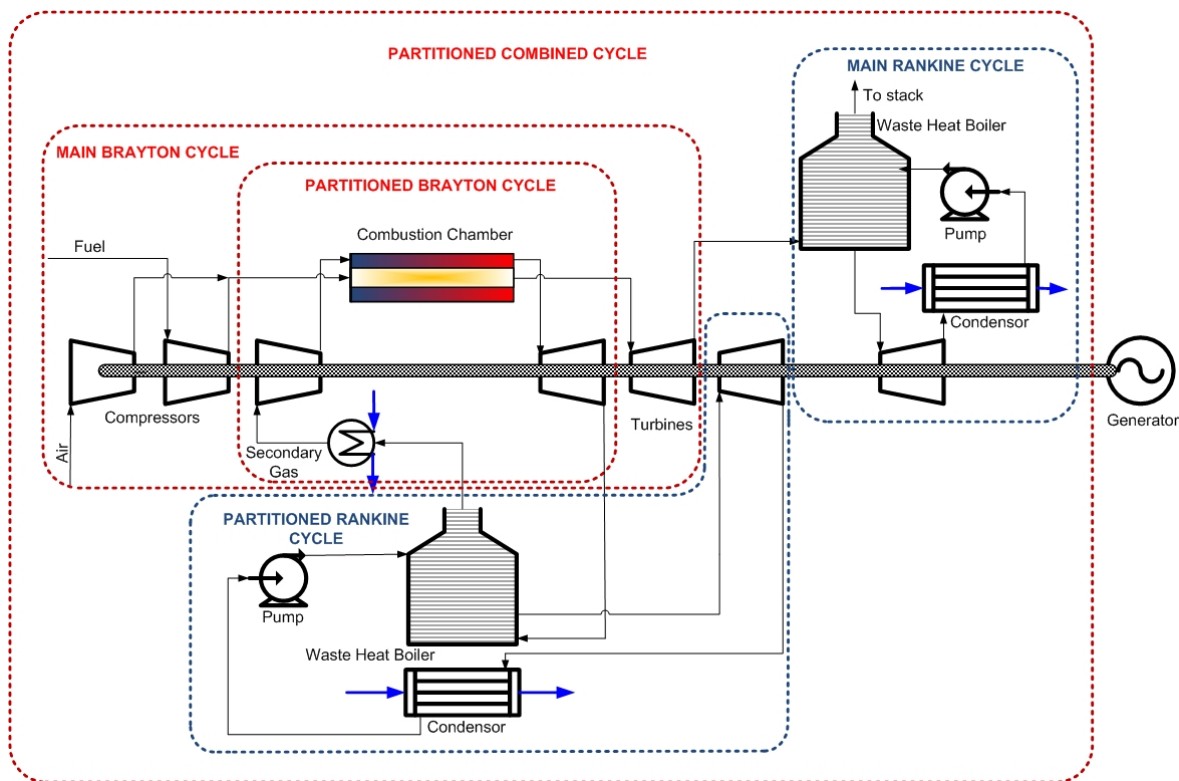

**Figure 2.** Novel proposed partitioned Brayton and Rankine cycles (PCGT) to efficiently recover cycle exhaust energy. Contrary to the main cycle, the partitioned Brayton cycle is a closed one.

## 3. Model Description

The GE 9HA.02 (General Electric, USA) combined cycle gas turbine is modeled and discussed in detail [54]. The errors in the model were dealt with by defining a range of parametric analysis, as shown in Table 1. Gas turbines have extensive moving machinery and offer a variety of loads. The flexibility in operation comes at a cost of efficiency loss. This decrease in efficacy is usually because of the process dynamics (discrepancy in mechanical losses is usually negligible because of the preventive maintenance schedules). Moreover, the ambient conditions also change frequently. Therefore, simulating such processes with fixed parameters may lead to erroneous conclusions. The best and worst process conditions were defined as upper and lower efficiency limits, respectively. Part load conditions can also be restrained in the same respective limits.

**Table 1.** Expected full load operating conditions range to confine the simulation errors and variations in real operating conditions.

| Parameters | Upper Limit | Mean Operating Conditions | Lower Limit |
|---|---|---|---|
| Gas Turbine Output (MW) | 577 | 564 | 551 |
| Air Intake (kg/h) $\times 10^{-6}$ | 3.342 | 3.51 | 3.695 |
| Fuel Intake (kg/h) $\times 10^{-3}$ | 89.57 | 92.93 | 96.650 |
| Gas Turbine Exhaust Temperature (°C) | 608 | 630 | 652 |
| Air Compressor work (MW) | 410 | 444 | 482 |
| Rankine cycle out Put (MW) | 178 | 191 | 204 |
| Overall Efficiency (%) | 61.8 | 59.57 | 57.27 |

The high pressure and temperatures in the process required a reasonable thermodynamic model to calculate the properties. Ideal gas assumption was therefore not appropriate. For the partitioned cycle five fluids were selected—air, argon, hydrogen, nitrogen, and carbon dioxide. The choice depended on

the range of thermodynamic properties. The Peng Robinson equation of state was used to estimate the properties. Other thermodynamic models may also be used, with expected discrepancies from Peng Robinson. As an example, for the Soave–Redlich–Kwong model, the maximum difference in compressibility factor for all the considered gases (at 30 bar and 25–1500 °C) was around ±1.7%. The validity of the calculation may be limited by the accuracy/applicability of the Peng–Robinson Equation of State (EoS). However, ASPEN HYSYS process simulator modified the critical properties to avoid substantial errors in the case of hydrogen and helium. Other quantities, like thermal conductivity and viscosity, were calculated by the polynomials regression based on the experimental data. According to our experience with industry, ASPEN HYSYS algorithms give reasonable accuracy for high pressure and temperature processes with hydrogen and carbon dioxide. Nevertheless, the results were accurate enough to draw a conclusion at least at qualitative level.

The compressor was intercooled by the help of jackets. A separate intercooler is usually not employed because of excessive pressure drops. The jacket/intercooler affect was inculcated in the model by defining the adiabatic efficiency. Hot spots in combustion chamber may produce excessive NOx. Therefore, their equilibrium amount in flue gases was considered. The waste heat boiler temperature approach must be adjusted to avoid excessive backpressure and a huge boiler size. This temperature approach was around ≈200 °C. However, depending on the overall heat transfer coefficients defined by different gases, this should be adjusted. Parameters were carefully selected to keep relative simulation errors the same for CCGT and PCGT.

*3.1. Heat Transfer and Efficiency Relationship*

Before the analysis of working fluids and equipment design, an optimization function was required to set parameters objectively. The Rankine cycle has an inherent latent heat efficiency loss. In a conventional CCGT, the higher the temperature and pressure of the gas entering the turbine, the higher the Brayton cycle/Rankine cycle efficiency. However, this does not necessarily result in higher CCGT overall efficiency. Equation (1) describes the relationship of overall CCGT efficiency ($\eta_{OA}$) as a function of Brayton cycle efficiency ($\eta_B$), Rankine cycle efficiency ($\eta_R$), heat exchanged in HRSG ($Q_{R,Ex}$), and fuel latent heat as input (IP). This relationship graphically shows a volcano type pressure ratio curve. Heat transfer in waste heat boiler between flue gasses and water/steam is a major factor, which is a function of flue gasses' outlet temperature (temperature approach), overall heat transfer coefficient, and exchanger area. The design of HRSG is therefore critical in maximizing the efficiency function.

$$\eta_{OA} = \eta_B + \eta_R\left(\frac{Q_{R,Ex}}{IP}\right) \tag{1}$$

A similar relationship for PCGT can be derived as Equation (2). Subscript 'p' indicates the quantities related to PCGT. In addition to partitioned HRSG heat transfer ($Q_{R,Ex,p}$), heat transfer in the combustion chamber ($Q_{C,Ex}$) also becomes significant. It is evident from this equation that if the Rankine cycle is not partitioned then the partitioned Brayton cycle efficiency ($\eta_{B,p}$) must be greater than the original Brayton cycle ($\eta_B$) to yield any benefit from the proposed scheme. However, as the exhaust temperature in PCGT can be lowered substantially (because of there being theoretically no water dew point), thus improved heat recovery in the partitioned Rankine cycle will positively affect the overall efficiency. The individual/local efficiencies are the function of temperatures and pressure ratios, which are directly affected by the heat exchangers. Therefore, any gain is directly related to the exchanger design. This argument is valid for almost any heat recuperation. However, the factual benefit of the proposed approach is the improvement in temperature gradient, which also reduces the pressure drop. Hence, Equation (2) may be used theoretically to maximize the efficiency, provided the entropy loss of the working fluid is comparable:

$$\eta_{OA,p} = \eta_B + \eta_R\left(\frac{Q_{R,Ex}}{IP}\right) + \left(\frac{Q_{C,Ex}}{IP}\right)(\eta_{B,P} - \eta_B) + \eta_{R,p}\left(\frac{Q_{R,Ex,p}}{IP}\right) \tag{2}$$

### 3.2. Pressure/Temperature Ratios for the Brayton Cycle

The CCGT pressure and temperature ratios are optimized in the original GE model GT. However, sensitization analysis is required for partition cycle. Figure 3 overall represents these results. The optimum pressure and temperature ratios were calculated using conditions bounded by the limits defined in Table 1. In Figure 3a the pressure ratio is plotted against Brayton cycle efficiency. In the Brayton Cycle, it is known that an optimum pressure and temperature ratio is required. However, high combustion temperature does not allow high pressures due to metallurgical limitations. Therefore, the pressure is not high enough to maximize heat to work conversion, resulting in high entropy (second law efficiency) loss. Any fluid that has less heat capacity than CCGT working fluid will inherently present the advantage of heat recovery. It is evident from the simulated data that $CO_2$-, Air-, $N_2$-, $H_2$-, and Ar-based partitioned Brayton cycles had maximum efficiency at pressure ratios of 1168, 105, 102, 79.7, and 19.9, respectively, at mean operating conditions. As the pressure ratio was enormously (impractically) high for these gases (except Ar), a separate partitioned Rankine cycle was required to make up the entropy loss because of the low turbine inlet pressure. However, for Ar the pressure ratio is achievable with comparable efficiency. This behavior of Ar is because of its lower heat capacity and compressibility factor. Similar behavior for temperature ratios can be inferred and is presented in Figure 3b. Exhaust temperatures of 372, 320, 319, 312, and 308 °C were calculated for $CO_2$, Air, $N_2$, $H_2$, and Ar, respectively, for the mean operating conditions pressure ratios. Inlet temperature to turbine was fixed at 1435 °C. These exhaust temperatures do not provide the optimum efficiency value for PCGT, as such low temperatures have an impact on HRSG heat transfer and thus the $Q_{R,Ex,p}$ in Equation (2) lowers the overall efficiency.

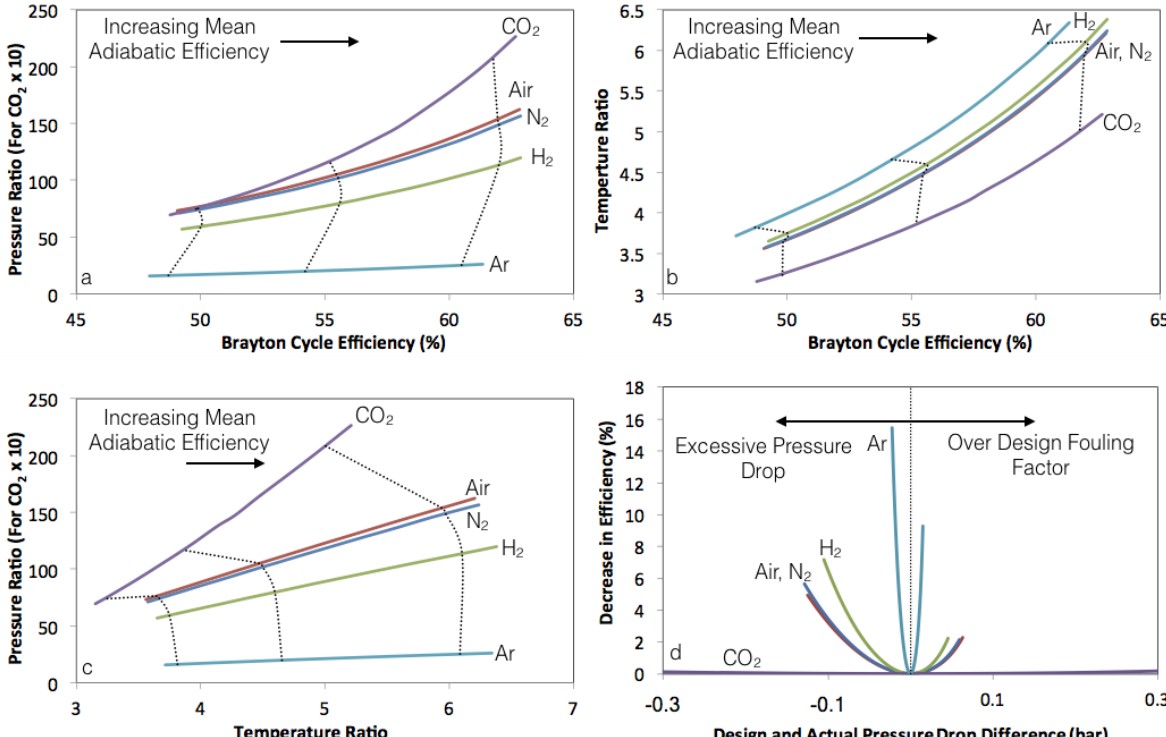

**Figure 3.** Light blue—Ar, green—$H_2$, blue—$N_2$, red—Air, purple—$CO_2$, (**a**) optimized pressure ratio versus simple cycle efficiency for various adiabatic efficiencies. Dotted lines represent the two limits and the mean operating conditions. Pressure ratio = turbine inlet Pressure/118 kPa. (**b**) Optimized temperature ratio versus simple cycle efficiency for various adiabatic efficiencies. Dotted lines represent the two limits and the mean operating conditions. Temperature ratio = 1435 °C/Exhaust Temperature. (**c**) Pressure ratio versus temperature ratio. (**d**) Differential of 'a', dotted line presents the zero x–axis.

The optimum pressure and temperature ratios for Brayton cycle are plotted in Figure 3c. The slope of these lines indirectly illustrates the entropy loss for various gases if the simple cycle is operated below the optimum pressure. The higher the slope, the higher the exergy cost will be. Thus, $CO_2$ was predicted to offer the highest loss. Figure 3d represents the differential of Figure 3a. These curves are important as they show the effect of excessive pressure drop in the downstream heat exchanger/boiler. The heat exchanger duty is a dynamic feature of fouling. Fouling factor is a design aspect. Although the waste heat recuperation is working on closed systems for partitioned cycle and aggressive fouling factors are not needed, a slight change in pressure drop may have drastic effect on cycle efficiency, as shown in the case of Ar. As an example, an efficiency drop of 3.2% was calculated for only 0.01 bar pressure drop greater than the designed pressure drop. Similar behavior was observed when Ar, as in the case of a new/cleaned heat exchanger, experienced low-pressure drop. This represents the negative effect of the low heat capacity gases. The effect will be much pronounced if the PCGT operates at part load conditions, where efficiency is expected to fall drastically. For the rest of the gases the affect is not much distinct. An adaptive controller for compressor vane adjustment may be used to counter this problem.

*3.3. Pressure/Temperature Ratios for the Rankine Cycle*

It must be noted that the partitioned Rankine cycle operating conditions are different from the main Rankine cycle mainly because of different temperature gradients. The overall efficiency depends on the rate of heat transfer to Rankine cycle and its efficiency. The temperature of the steam must be kept above 325 °C, or preferably above the critical temperature (374 °C). At temperatures below 325 °C the optimum pressure is greater than the saturation pressure, thus seriously compromising the steam turbine efficiency and possible high condensation in turbine exhaust. Although screw expanders are reported to safely handle a greater degree of sub-saturation [59,60], for large turbines the volume of steam is too large to be economically depressurized in these expanders. Higher temperatures are preferable as both the heat transferred and the Carnot efficiency increases. However, the maximum steam temperature is dictated by the practical limitations of the waste heat boiler surface area and pressure drop. The local Rankine cycle efficiency maxima are shown in the Figure 4. In order to provide maximum advantage, the turbine exit pressure was maintained at 10 kPa. The figure can be generated at different overall adiabatic efficiencies. The optimum pressure curves can be regressed to the resultant Equation (3), where P and T are in 'kPa' and 'K', respectively. However, the water cannot be pressurized to optimum value as it results in excessive condensation. So, it is adjusted to keep the steam quality above 98% before the condenser.

$$P = \left[0.0153\eta_A^2 - 3.0128\eta_A + 420.18\right]exp(0.0064T) \quad (Regression\ Error\ \pm 0.54\%) \tag{3}$$

The overall efficiency equation is directly proportional to heat transfer from exhaust turbine gases in HRSG and does not specifically represent the maximum achievable temperature as in the case of the Carnot cycle. Therefore, HRSG design limitations will largely dictate the optimum point. High steam temperature reduces the heat transfer rate and low steam temperature reduces the turbine efficiency for a HRSG with fixed heat transfer area. Therefore, there is a trade-off. As previously described, in PCGT partitioned working fluid temperature can be reduced much lower than the CCGT fluid, thus the same HRSG (same area) is used to transfer heat between partitioned fluid and steam, as simulated in CCGT. However, the area of HRSG used to exchange heat between main exhaust gases and steam in PCGT is smaller with same ratio to keep the exhaust temperature above 171 °C.

Both the pressure and temperature ratio of the individual Brayton cycle and Rankine cycle in CCGT and PCGT have their local efficiency maxima. Care must be taken to avoid simulation at these conditions as the overall efficiency has a global maximum value at other different conditions.

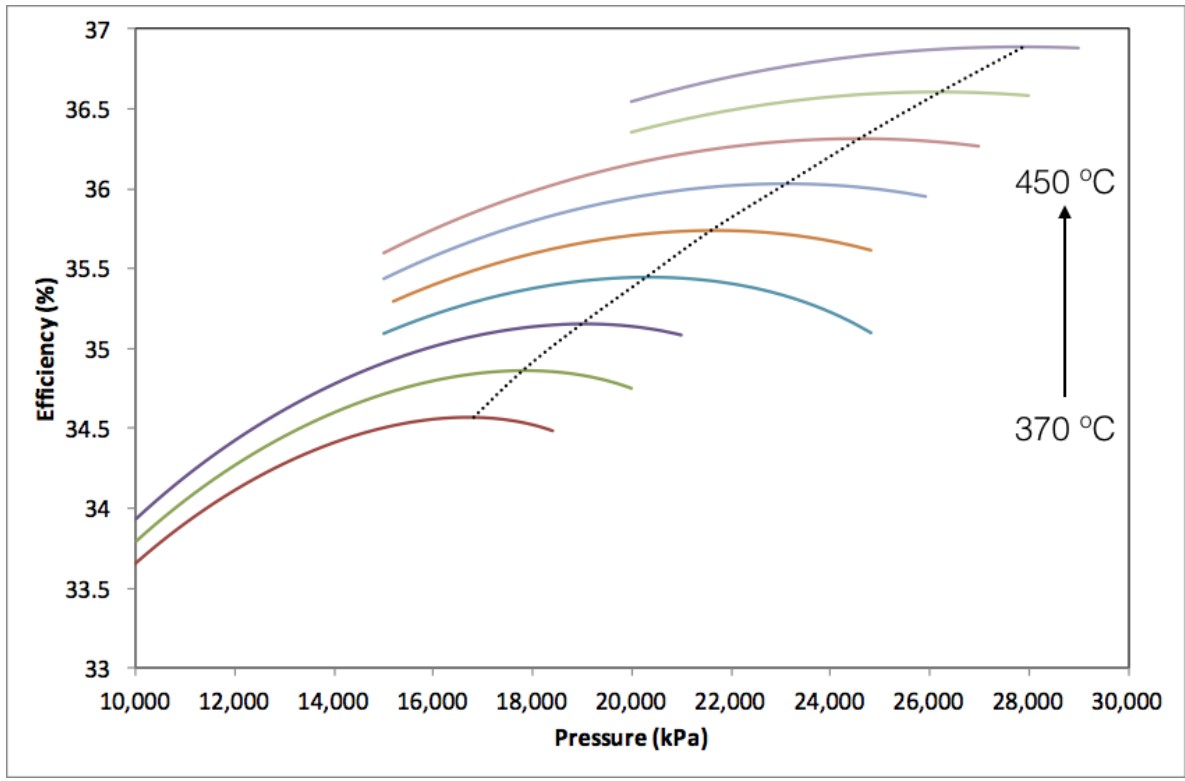

**Figure 4.** Efficiency versus pressure for the Rankine cycle at temperatures from (370–450 °C), with adiabatic efficiency of 89.9%. The turbine exit pressure is 10 kPa. Dotted line represents the optimum pressure.

### 3.4. Heat Transfer Rates in Combustion Chamber

There are 16 Dry Low $NO_x$ (DLN) type combustion chambers being used in H-class series turbines. These chambers are designed so that all the compressed air does not pass through the chambers directly and most of the air is diffused to cool the flame adiabatic temperature. However, in the proposed PCGT all of the air must pass through the chamber. Excess air of 15% was allowed to ensure complete combustion. Furthermore, the PCGT partitioned fluid must cool the flame indirectly. Therefore, a shell and tube heat exchanger replaced each combustion chamber. Flow was equally divided. Average velocities of the gases were maintained at 25 m/s. The velocities of the gases were maintained to ensure the momentum heat transfer within the combustion and partitioned zone. Flame length must be calculated to determine the heat release pattern in the combustor tube. Many factors influence its length and Computational Fluid Dynamics (CFD) calculations are required to exactly solve the 3D heat balance equation. However, using a 1D approach in heat transfer with an estimation of the flame length may provide the dimensions/heat distribution in the combustor and partitioned fluid space within a reasonable error range (well-stirred lumped model cannot provide a rational estimate of the required length of combustor chambers due to a high adiabatic temperature).

Figure 5 shows the algorithm used for calculations. The main ($T_{m,i}$) and partitioned fluid inlet temperatures ($T_{p,i}$) were obtained by the compression calculations with different adiabatic efficiencies discussed earlier. The main fluid outlet temperature ($T_{m,o}$) was fixed at 1435 °C by manipulating the partitioned fluid flow rate ($m_p$). The main fluid flow rate ($m_m$) is adjusted to keep the gas turbine output at 755 MW. The partitioned fluid outlet temperature ($T_{p,o}$) was obtained by an iterative convergence of the problem. Complete heat transfer was considered at the seed value (1435 °C). The heat transfer and heat of reaction were balanced with temperatures by adjusting the flow rates before any further calculations. This exercise was essential to obtain the boundary conditions. In the 1D model (plug flow, Equation (4)), rapid combustion in a differential length (temperature) was considered and the mixture

achieved the partial adiabatic temperature instantaneously with no radial heat distribution. Only convective heat transfer was assumed to hold across the surface (although, radiation is important on hot side but it is not considered to give maximum disadvantage to the PCGT in order to reduce model errors). The solution was obtained by a code written in MATLAB. The step length was adjusted to keep the error less than $\Delta T < 0.1°C$ for both partitioned and main fluid. As the mixture was changing both chemically and physically because of the combustion reaction and temperature, respectively, the properties in each differential length were calculated using the thermodynamic model as used in ASPEN HYSYS. The information was then transferred to MATLAB for each discretized step.

$$\frac{dq}{dL} = Q_{generated} - Q_{Exchanged} \tag{4}$$

One-dimensional flame length for various fuels can be estimated by the ratio '$F_L/d_j$', where '$F_L$' is the flame length and $d_j$ is the jet injection diameter. A range of '$F_L/d_j$' ratio can be found for different fuels. A value of '$F_L/d_j \approx 200$' for natural gas is usually reported [61]. Furthermore, the effect of swirl must be accounted for at high velocities. Various formulas are reported in the literature to account for the swirl. A moderate factor of $(1 + 2 S)$ in flame length reduction was thus used [62,63], where 'S' is the swirl number. Once the flame length is calculated then boundaries for the heat balance equations can be defined, as it is assumed that all the heat released by chemical reaction/combustion is in the flame and the rest of the chamber length only acts like a simple heat exchanger. The heat dissipation within the flame length can be estimated by an exponential decay function, as provided in Equation (5), where '$q(L)$' is the heat released at any flame length and '$Q_o$' is the total heat released. Partitioned fluid temperature thus measured updated the solution and the flow rates adjustment algorithm was repeated before Equation (4) was solved with new values. Tolerance limits of 0.1 °C and 10 kW were chosen for partitioned and main fluids flow adjustment.

$$q(L) = \frac{4.6\, Q_o}{F_L} exp\left[\frac{-4.6L}{F_L}\right] \tag{5}$$

$$Nu = 0.023\, Re^{0.8} \times Pr^{0.4} \tag{6}$$

The Dittus–Boelter equation (Equation (6)) was used to estimate local heat transfer coefficients in each differential length. The Prandtl number, Reynold number, and L/D ratio were within the specified range of 0.6 to 16, >10,000, and >10, respectively. The log mean temperature difference (LMTD) was taken equal to adiabatic temperature and partitioned fluid inlet temperature difference. This introduced an error in the model and was dealt with by decreasing the step size, causing a maximum temperature error of 0.04 °C. Combined with the heat capacity balance equation, Equation (4) was thus solved. The exchanger total length was fixed to 7 m with 100 tubes. The pressure drop estimation details can be found in the relevant text [64]. Pressure drop can have a drastic effect in some fluids, as discussed earlier. As the output drops because of the pressure drop, if the change in the gas turbine output is more than 10 kW the whole algorithm restarts by adjusting the main fluid (air + fuel) mass flow rates.

As the Rankine cycle was chosen as a bottoming cycle in CCGT, the partitioned cycle Rankine cycle conditions must be defined before presenting any comparison. The pressure of the Rankine cycle was adjusted (according to the temperature) as described in previous section. However, the optimum temperature was required. To completely harness the advantage of cycle partitioning, HRSG must be designed to keep the main fluid outlet temperature as low as possible. The HRSG area was kept same as that for the main cycle. However, the flow rates were lesser, if not halved, compared to the original cycle. In order to keep the overall heat transfer coefficient in the same range, the velocities (Reynolds number) of the fluids were adjusted. Thus, after fixing 'U' and 'A', sensitivity analysis was conducted on temperature with respect to overall efficiency.

**Combustion Chamber Simulation Algorithm**

Partitioned fluid inlet Temperature
$T_{p,i}$ = HYSYS calculation after compression
Main fluid inlet Tremperature
$T_{m,i}$ = HYSYS calculation after compression
Main fluid outlet Temperature, $T_{m,o}$ = 1435 $^{o}$C
Partitioned fluid outlet temperature $T_{p,o}$ = 1435 $^{o}$C
(Seed Value)

Seed Value of $\dot{m}_m$ and $\dot{m}_p$

$\dot{m}_m$ and $\dot{m}_p$

Dimension of the heat exchanger/combustor adjusted to keep the inlet velocities at 25 m/s
Total length adjusted to 7 m

HYSYS iterative procedure:
Cp values averaged at inlet and outlet temperatures
$Q_m = \dot{m}_m Cp(T_{m,o} - T_{m,i})$
$Q_p = \dot{m}_p Cp(T_{p,o} - T_{p,i})$
ΔQ reaction at $T_{m,o}$ calculated

MATLAB Code:
Flame length estimated by FL/di = 200 with swirling
Step length 'dL' chosen with tolerance limit for stiffness 0.1 $^{o}$C in temperatures of both partitioned and main fluid temperatures

Average heat capacity, viscosity, density and thermal conductivity of hot and cold fluid with in the differential length calculated by Peng-Robinson property package using HYSYS

ΔQ reaction slightly adjusted to compensate the exchange error between HYSYS and MATLAB by keeping main fluid temperature at 1435 $^{o}$C

'Eq 5' is used to calculate Qgenerated in each 'dL' uptil flame length
Overall heat transfer coefficient evaluated by Dittus-Boelter eq.
Qexchanged is evaluated by taking LMTD = ΔT

Eq.4 is then discretized and solved over entire length
Partitioned fluid outlet temperature is measured

Adjust the value of $\dot{m}_m$

Adjust the value of $\dot{m}_p$

Output = 755 W
Tolerance = 10 kW
　Yes　　　No

ΔQ reaction = $Q_m + Q_p$?
Tolerance = 10 kW
　No　　　Yes

Adjust the value of $T_{p,o}(n)$ to new value
　No　　　$T_{p,o}(n+1) = T_{p,o}(n)$
Tolerance = 1 $^{o}$C
　yes

Calculate the pressure drop
Calculate the overall output

Output = 755 W
Tolerance = 10 kW
　No　　　Yes

Assumptions for each discretized length:
1. Properties of the fluids are averaged at fluid inlet and outlet conditions
2. Flame adiabatic temperature is instantly achieved
3. Negligible Heat losses

Exit and record values

The algorithm is repeated for different temperatures of the outlet steam from HRSG. The case study option in ASPEN HYSYS is used to determine the optimum temperature (= 5 $^{o}$C) with respect to overall efficiency.

**Figure 5.** Combustor chamber design and HRSG convergence algorithm. Discretized length = 0.001 m. $T_{p,i}$, $T_{p,o}$, $m_P$ = inlet temperature, outlet temperature, and mass flow rate of partitioned fluid, respectively. $T_{m,i}$, $T_{m,o}$, $m_m$ = inlet temperature, outlet temperature, and mass flow rate of main fluid, respectively. $Q_m$ and $Q_p$ are the heat absorbed or liberated by the main and partitioned fluid, respectively. $F_L$ = flame length. $d_j$ = jet injection diameter. dL = step length. Seed values for the algorithm were obtained by equating the mass flow rates in heat capacity equation using 1435 °C as the partitioned fluid outlet temperature.



## 4. Result and Discussion

### 4.1. Temperature and Viscosity Profiles

Figure 6 presents the temperature profiles of the main and partitioned fluid in the combustion chamber. The distance between the green (maximum temperature gradient) and red (temperature gradient at highest main fluid temperature) vertical lines is a function of heat transfer and flame length. The higher the distance the better the heat transfer. As expected, the behavior of nitrogen was not very different from air (nitrogen is a main constituent of air). However, the highest temperature tended to increase. There were two main factors that controlled this temperature: fluid conductivity and the amount of fuel burned. In the case of nitrogen, the difference was because of the conductivity. Carbon dioxide showed the best heat transfer, despite the lower maximum temperature gradient. Another fact that favors carbon dioxide is its lower entering temperature because of its better compression efficiency. This is due to its higher molecular weight and density. Hydrogen showed a similar temperature profile but its pressure drop was lowest among the gases studied (Figure 7, provides the viscosity profiles). Furthermore, hydrogen has previously used as heat transfer medium in electrical generators but was gradually replaced by helium due to safety concerns. The behavior of helium is expected to be not much different because of almost same viscosities as those of hydrogen under the studied conditions. On the other hand, argon not only provided a huge pressure drop but also showed poor heat transfer properties. The pinch temperature remained 42 °C at 7 m of the exchanger length, the highest among other fluids. Flame length was a direct measure of the load being shared between the main and partitioned cycle, as more load on the main cycle injects more fuel and thus a higher flame length. The heat duties of the combustor for hydrogen, air, and nitrogen at different adiabatic efficiencies were comparable. As the adiabatic efficiency of the cycle components increased, it decreased the irreversible losses and thus the heat input. A comparison between the heat duties of carbon dioxide and argon provides a noteworthy observation. Argon duty was lower than carbon dioxide duty at higher adiabatic efficiencies but the reverse is true at lower adiabatic efficiencies. This is linked to the optimization algorithm based on Equation (2). At high adiabatic efficiencies, the argon Brayton cycle became more effective and took higher loads than carbon dioxide, while its pressure drop remained the same. The entropy loss because of carbon dioxide thus became higher than the combined losses (entropy and pressure drop) offered by the argon. This resulted in a better overall performance by argon at higher components of adiabatic efficiencies. The average temperatures after the combustor were 1429 °C, 1428 °C, 1429 °C, 1393 °C, and 1433 °C for nitrogen, hydrogen, air, argon, and carbon dioxide, respectively.

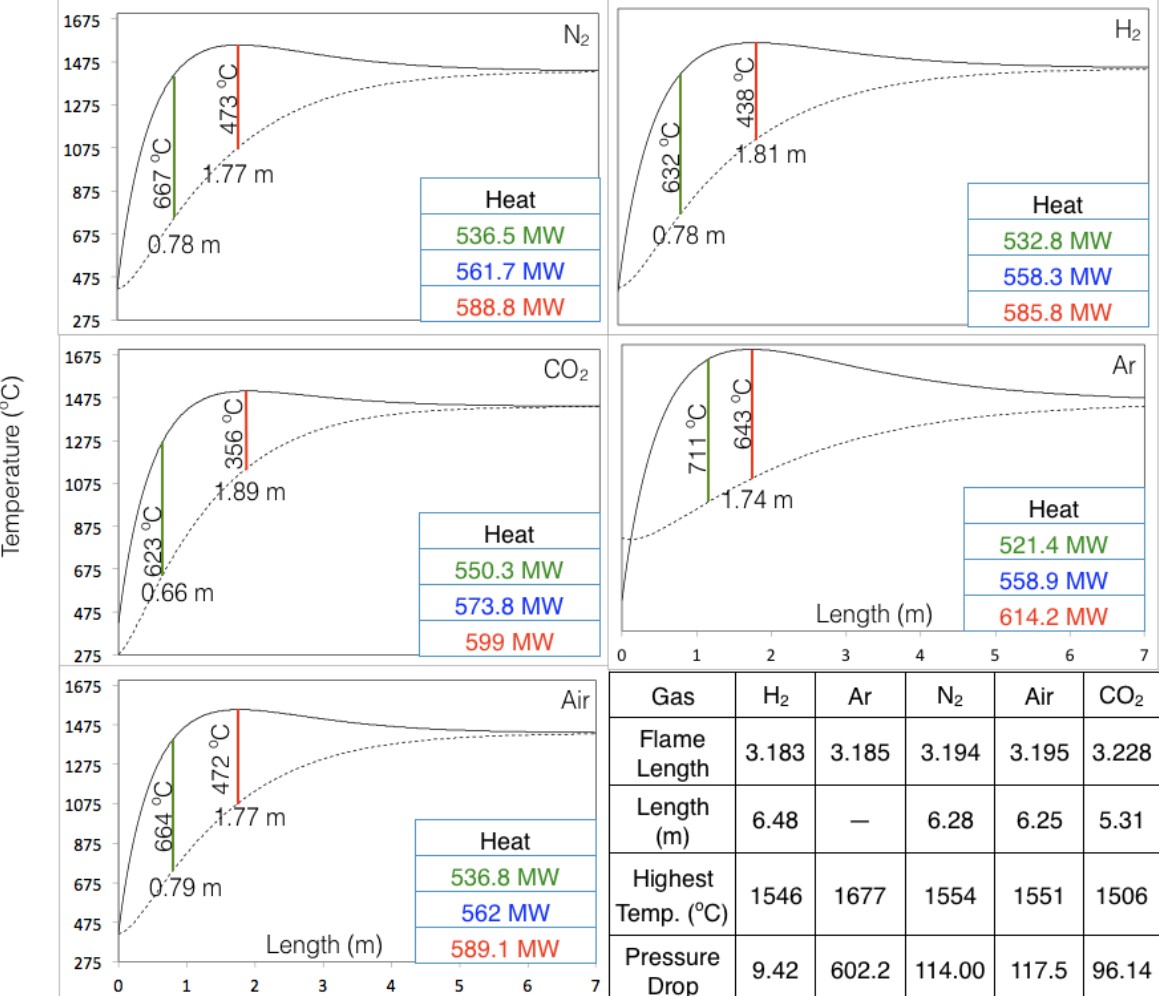

**Figure 6.** Temperature profiles in the combustor/heat exchanger at a mean adiabatic efficiency of 92.7%. Solid and dotted lines represent the main and partitioned fluid temperatures as a function of length, respectively, with a maximum error ±1 °C. Green and red lines represent the highest and maximum temperature gradient, respectively. Inset tables present the heat exchanged in the combustor; green, blue, and red represent the adiabatic efficiencies of 95.63%, 92.7%, and 89.9%, respectively.

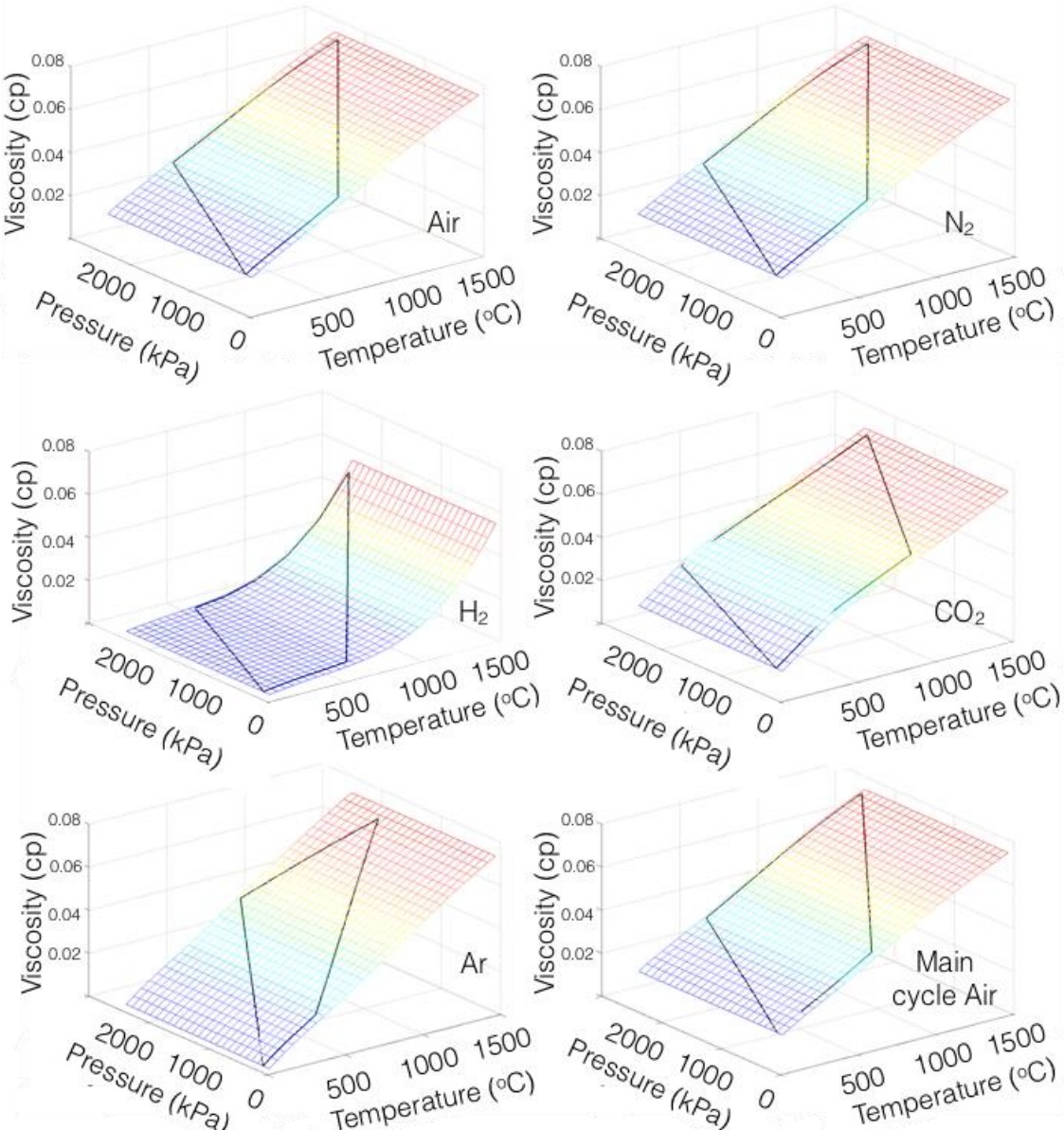

**Figure 7.** Viscosity change for different fluids shown as 3D surface plots. Black lines represent the viscosity profiles of the partitioned Brayton cycle at 92.7% adiabatic efficiency.

## 4.2. Heat Capacity Rates

Heat capacity rate data are provided in Figure 8. As per the surface plots, the heat capacities of argon, air, nitrogen, carbon dioxide, and hydrogen at the turbine inlet were 0.5229 kJ/kg °C$^{-1}$, 1.228 kJ/kg °C$^{-1}$, 1.255 kJ/ kg °C$^{-1}$, 1.365 kJ/ kg °C$^{-1}$, and 16.27 kJ/ kg °C$^{-1}$, respectively. Higher heat capacity translates into lower flow rates and thus low volumes to be handled by individual GT components. If the heat capacity value is high, higher capacity rates are available before the turbine, and thus more power can be extracted. Such numbers are interpreted by calculating the slopes of the Brayton cycle lines with respect to temperature and pressure. The conclusion can be generalized from Table 2. Lesser slopes 'dCp/dT' and 'dCp/dP' in the compressor than in the turbine are beneficial to the overall efficiency of the gas turbine. This provides an advantage in better heat/work recovery, although the difference in slopes is most suitable in the case of argon but its high viscosity results in higher-pressure drops. Furthermore, very low heat capacity rates cause very low temperatures at the Brayton cycle turbine outlet. This then requires extra combustion for the Rankine cycle to

work, causing an abrupt drop in combined cycle overall efficiency (Equation (2)). On the other hand, hydrogen promises less pressure drop as well as a considerable heat capacity advantage and suitability for Rankine cycle workability. Nitrogen is better than air but carbon dioxide offers the worst-case scenario. The same also translates to the heat capacity rates for different gasses, as calculated by the optimization algorithm. The Prandtl number also becomes small with lower heat capacity values, and thus it requires a large exchanger/combustor area to avoid a Carnot efficiency penalty, as in the case of argon.

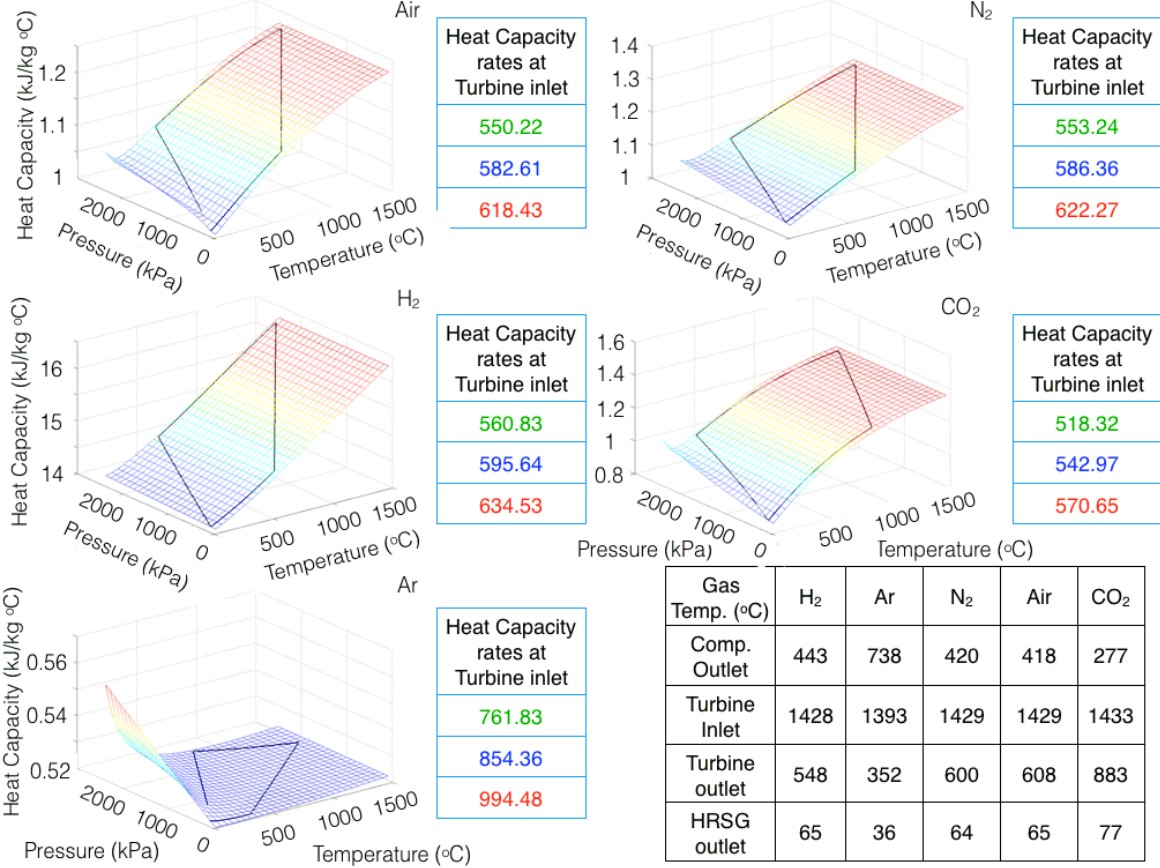

**Figure 8.** Heat capacity data for different fluids shown as 3D surface plots. Black lines represent the heat capacity profiles of the partitioned Brayton cycle at 92.7% adiabatic efficiency. Adjacent tables provide the heat capacity rates. Green, blue, and red represent these rates at adiabatic efficiencies of 95.63%, 92.7%, and 89.9%, respectively. The main table provides the mean temperatures at different locations of the partitioned cycle ($\eta$ = 92.7%).

Carbon dioxide offers high heat transfer in the combustion chamber, but its heat capacity drastically drops with a decrease in pressure. Thus, it provides the poorest heat transfer in the HRSG. This offers a considerable draw back. In the literature, carbon dioxide is extensively used at super critical conditions to avoid this.

**Table 2.** Heat capacity slopes with respect to temperature and pressure in the compressor and turbine for different gases and their proportional differences.

| Gas | Argon | | Nitrogen | | Hydrogen | | Carbon Dioxide | | Air | |
|---|---|---|---|---|---|---|---|---|---|---|
| Equipment | Compressor | Turbine | Compressor | Turbine | Compressor | Turbine | Compressor | Turbine | Compressor | Turbine |
| $dCp/dT$ ($\times 10^{-4}$) | 0.0103 | 0.0221 | 1.92 | 1.35 | 13.7 | 16.4 | 7.29 | 1.75 | 2.17 | 1.24 |
| $dCp/dP$ ($\times 10^{-4}$) | 0.00209 | 0.0139 | 0.485 | 0.522 | 3.96 | 6.39 | 1.28 | 0.444 | 0.542 | 0.476 |
| Proportional Difference (w.r.t T) | 1.16 | | −0.29 | | 0.20 | | −0.76 | | −0.43 | |
| Proportional Difference (w.r.t P) | 5.64 | | 0.08 | | 0.61 | | −0.65 | | −0.12 | |

### 4.3. Entropy (Second Law) and Density Profiles

Entropy and density profiles are shown in Figures 9 and 10, respectively. As per the volumetric data, carbon dioxide offers the least volume. Hydrogen is comparable to air and nitrogen. Argon offers huge volumes due to its high flow rates (caused by low heat capacity), thus unacceptably increases the equipment sizes.

Generally, more loads shared by PCGT increase the overall efficiency. This is because the exhaust heat is recovered (T < 171 °C), which compensates for the work lost due to the pressure drop and irreversibility associated with the proposed scheme. Reversible adiabatic efficiency of 100% has zero entropy change. However, if variable irreversibility is accounted, gases started to behave differently under different conditions. Second law inefficiencies, as shown in the adjacent tables to entropy profiles, clearly favor carbon dioxide as partitioned fluid. However, it does not offer much benefit at higher adiabatic efficiencies and the cycle become inefficient (the reason is stated in the previous section). Carbon dioxide starts to offer a positive efficiency increase once the cycle adiabatic efficiencies start to drop below ≈90.5%. On the other hand, argon pressure losses surpass the break, even at adiabatic efficiencies above ≈94%. Based on the discussion, hydrogen offers the best choice as partitioned fluid. It must be noted that helium properties also promise the same conclusion.

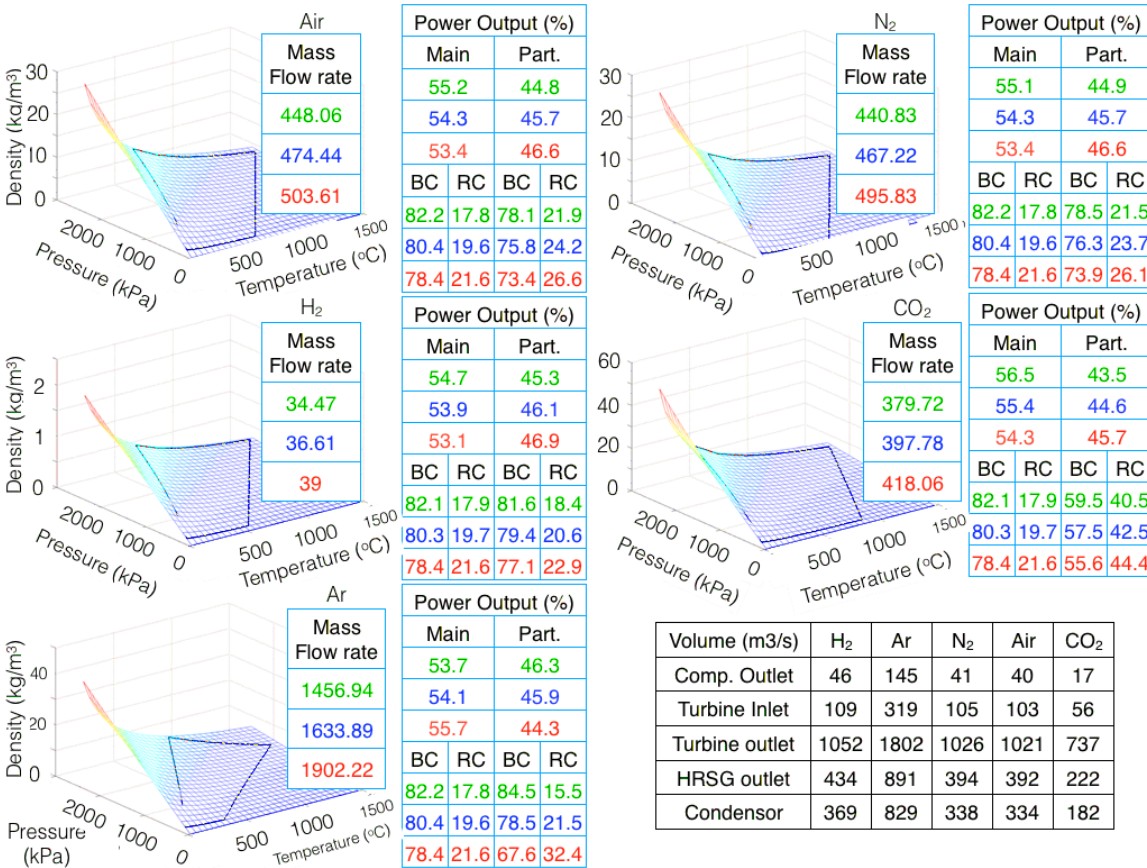

**Figure 9.** Density data for different fluids shown as 3D surface plots. Black lines represent the entropy profiles of the partitioned Brayton cycle at 92.7% adiabatic efficiency. Adjacent tables provide the mass flow rates and load distribution between the main, partitioned cycle, Brayton cycle, and Rankine cycle. Green, blue, and red represent these losses at adiabatic efficiencies of 95.63%, 92.7%, and 89.9%, respectively. The main table provides the volumetric flow rates at different locations of partitioned cycle (η = 92.7%).

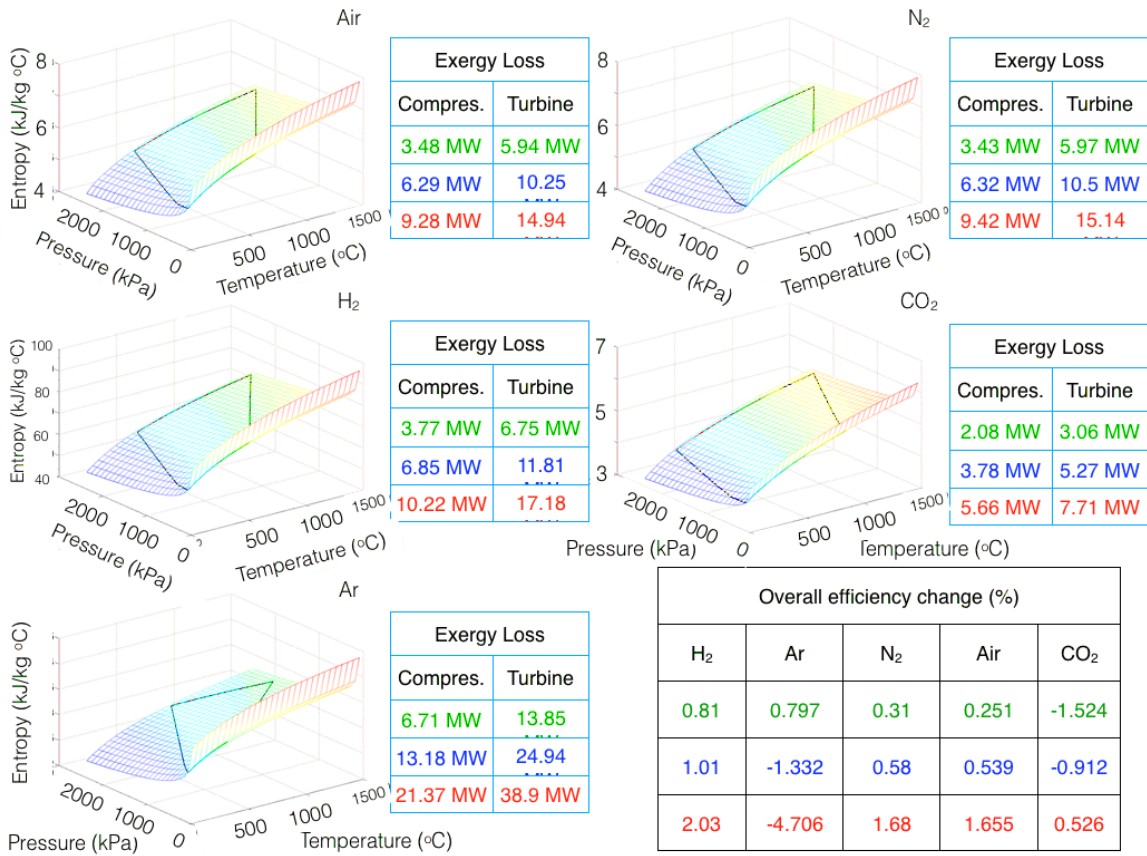

**Figure 10.** Entropy data for different fluids shown as 3D surface plots. Black lines represent the entropy profiles of the partitioned Brayton cycle at 92.7% adiabatic efficiency. Red and blue numbers represent the slope of the curves with respect to temperature and pressure, respectively. Adjacent tables provide the exergy loss in the compressor and turbine. Green, blue, and red represent these losses at adiabatic efficiencies of 95.63%, 92.7%, and 89.9%, respectively. The main table provides the overall efficiency change by the addition of partitioned cycle.

## 5. Conclusions

In this study, heat recovery from the flue gases in CCGT was maximized. As the temperature of the flue gases cannot be dropped below a certain limit, the original cycle was partitioned to partially cool the clean exhaust in PCGT.

The model was simulated on ASPEN HYSYS and MATLAB. CCGT and PCGT unit operations were modeled using the standard modules in ASPEN HYSYS. Before simulation, initial sensitivity analysis was done for the Brayton cycle and Rankine cycle to assist the PCGT optimization algorithm. The design calculations for the combustor/heat exchanger were done in MATLAB. Care was taken to share the fluid properties between the two softwares with decent tolerance limits and at every discretized step. The error compensation was well accounted for to simulate the model (both in ASPEN HYSYS and MATLAB). Thus, the results confidence level was high. All the inputs (fuel, inefficiencies, pressure drops, heat losses, and cooler Coefficient of Performance (COP)) are accounted for to rationalize the conclusion.

Five different gasses were studied. The minimum exhaust temperature (heat recovery) was recorded for argon. However, its high viscosity (pressure drop), extra burners/combustion for the Rankine cycle, and high Carnot cycle penalty (due to less heat exchange) make its use possible only at high cycle adiabatic efficiencies. Furthermore, its pressure/pressure drop control is crucial to harness any extra benefit from PCGT. Carbon dioxide offered moderate pressure drops but low heat to work conversion efficiency and lower heat transfer in HRSG, making use possible only at lower adiabatic

cycle efficiencies. Nitrogen and air promised good heat recovery and comparable pressure drops. However, hydrogen was the best candidate for the proposed cycle because of its better properties at PCGT conditions. Quantitatively, the average heat capacity of any tailored gas should be lower in the compressor than the turbine, but its absolute value should not be too low, to decrease the exhaust turbine temperature below any practical limit of Rankine cycle (HRSG). Viscosity should be low and the fluid should offer entropy losses lower than air. Supercritical $CO_2$ is being studied extensively but high pressure is also associated with this fluid. On the other hand, helium and hydrogen must be given their due consideration, as these offer better properties at low pressures and thus higher efficiencies.

**Author Contributions:** A.A.T. and M.E.S. conceived the idea and did the programming including validation. The results were analyzed and discussed with S.S.A.A. who also helped in improving the text.

**Funding:** No funding was acquired for this work.

**Conflicts of Interest:** The authors declare no conflict of interest.

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
