# Peer review of "Thermodynamic Analysis of Partitioned Combined Cycle using Simple Gases"

_applsci, doi:10.3390/app9194190_

Round 1

Reviewer 1 Report

line 216: Better say: Hence, the maximization of efficiency given by Equation 2 …

line 344: Please provide Dittus-Boelter equation with all parameter and their values that were used

Equation 4: Dimensions of right hand side terms seem to be different from those corresponding to energy/length. Please check

Figure 5: Please specify consistently whether the algorithm is for simulation or optimization; in case of optimization, you need to provide the specifics of the optimization method used. In the figure, specify how the seed values were obtained, and how they were adjusted

Abstract and Conclusions need to be made significantly more consistent. The objective on line 469 is too late for mention in the paper

Lines 494-497: These should be moved out of Conclusion as they do not carry what was done in this study

Lines 22-23: For the outcome mentioned in these lines, please provide a separate section in Results and Discussions, and use a table to rank the gases based on different thermodynamic properties

Abstract: Either remove the last sentence or make it more specific 

Authors need to comment on whether and how the results would vary using another equation of state, e.g., Soave Redlich Kwong equation of state

Author Response

Dear Reviewer

Thank you for reviewing the article my point wise reply is as follow:

line 216: Better say: Hence, the maximization of efficiency given by Equation 2 …

Reply: The sentence is changed to "Hence, Equation 2 may be used to theoretically maximize the efficiency."

line 344: Please provide Dittus-Boelter equation with all parameter and their values that were used

Reply: Equation is provided along with a range of parameters used

Equation 4: Dimensions of right hand side terms seem to be different from those corresponding to energy/length. Please check

Reply: Thank you for pointing out the typing error. q represents the flux and Q represents heat generated/transferred in the controlled volume with units J/m3.

Figure 5: Please specify consistently whether the algorithm is for simulation or optimization; in case of optimization, you need to provide the specifics of the optimization method used. In the figure, specify how the seed values were obtained, and how they were adjusted

Reply: It is not the optimization in true sense, so the caption is changed. Seed values for the algorithm are obtained by equating the mass flow rates in heat capacity equation using 1435oC as partitioned fluid outlet temperature

Abstract and Conclusions need to be made significantly more consistent. The objective on line 469 is too late for mention in the paper

Reply: It is changed accordingly

Lines 494-497: These should be moved out of Conclusion as they do not carry what was done in this study

Reply: done

Lines 22-23: For the outcome mentioned in these lines, please provide a separate section in Results and Discussions, and use a table to rank the gases based on different thermodynamic properties

Reply: I must admit that there was an ambiguity in writing these lines. These are modified and the values of the ranges can be deduced from the tables in  figures 7, 8, 9 and 10. Providing them separately may confuse the reader.

Abstract: Either remove the last sentence or make it more specific 

Reply: Sentence is removed.

Authors need to comment on whether and how the results would vary using another equation of state, e.g., Soave Redlich Kwong equation of state

Comments are added line 187-195

Best regards

Reviewer 2 Report

Although the manuscript is interesting, there are several problems, including one major and a few minor ones, therefore I would recommend major revision.

MAJOR ISSUES

This is a detailed case study and it should be very accurate. The calculation is good, but using Peng-Robinson equation of state for air, hydrogen or CO2 is not correct. For helium, it can be used in this temperature range. This equation of sate can be used only for nitrogen or argon (or some lower alkanes) in some extent. For all other materials, the results will be only qualitative. For CO2, air or hydrogen, there are very accurate reference equation of states (for example the Wagner-Spann for CO2). The Authors should mention, that the validity of their calculation is limited by the accuracy/applicability of the Peng-Robinson EoS. Having a fast comparison for C2 with Peng-Robinson and Wagner-Span EoS, for density, the error of Peng Robinson is around 0.5%. Concerning similar errors for other quantities, the Authors should be very careful by giving the Power outputs with 0.1 % accuracy.

MINOR ISSUES

English should be improved, sometimes it is not understandable (like in line 157, „…better lesser entropy…” ). Numbers and axis labels are hardly readable for Figure 9, but in some extent, also for Figures 7 and 8. 5 is a very important figure, but the letters are quite blurry, should be sharper. In some cases, the Figure and its first mention are quite far (like Fig 6, mentioned first on p12, while the figure is sown only two pages later). Using only the abbreviations is a bit too colloquial; sometimes reader might e lost, therefore the Authors should use Rankine Cycle instead of RC, combined cycle gas turbine, instead of CCGT, etc, at least for important sentences. In Conclusions, writing “…initial sensitivity analysis done for BC and RC to assist PCGT optimization algorithm” would force the reader to go back to the beginning to decode the abbreviations. Also, it might be a god solution to have a list of abbreviations.

Author Response

Dear Reviewer

Thank you for reviewing the article my reply is as follow:

This is a detailed case study and it should be very accurate. The calculation is good, but using Peng-Robinson equation of state for air, hydrogen or CO2 is not correct. For helium, it can be used in this temperature range. This equation of sate can be used only for nitrogen or argon (or some lower alkanes) in some extent. For all other materials, the results will be only qualitative. For CO2, air or hydrogen, there are very accurate reference equation of states (for example the Wagner-Spann for CO2). The Authors should mention, that the validity of their calculation is limited by the accuracy/applicability of the Peng-Robinson EoS. Having a fast comparison for C2 with Peng-Robinson and Wagner-Span EoS, for density, the error of Peng Robinson is around 0.5%. Concerning similar errors for other quantities, the Authors should be very careful by giving the Power outputs with 0.1 % accuracy.

Reply: The thermodynamic properties were calculated by using HYSYS. In our previous work using PR EOS for Carbon dioxide gives results very close to the industrial data. Ref: "Aqeel Ahmad Taimoor, Saad Al-Shahrani, & Ayyaz Muhammad, “Ionic Liquid (1-Butyl-3-Metylimidazolium Methane Sulphonate) Corrosion and Energy Analysis for High Pressure CO2 Absorption Process.” Processes, 6(5) 2018, 45." For hydrogen, HYSYS uses modified critical properties for estimation and many industrial simulations can be found by using the same method. I should have mentioned this in the article. But as you have pointed, I added the information at line 187-195

English should be improved, sometimes it is not understandable (like in line 157, „…better lesser entropy…” ). Numbers and axis labels are hardly readable for Figure 9, but in some extent, also for Figures 7 and 8. 5 is a very important figure, but the letters are quite blurry, should be sharper. 

Reply: The Matlab graphs are quite low in quality. The values are added manually in figures 7 - 10. English is also improved in various sentences

In some cases, the Figure and its first mention are quite far (like Fig 6, mentioned first on p12, while the figure is sown only two pages later). 

Reply: This will be addressed after the final typeset of the article.

Using only the abbreviations is a bit too colloquial; sometimes reader might e lost, therefore the Authors should use Rankine Cycle instead of RC, combined cycle gas turbine, instead of CCGT, etc, at least for important sentences. In Conclusions, writing “…initial sensitivity analysis done for BC and RC to assist PCGT optimization algorithm” would force the reader to go back to the beginning to decode the abbreviations. Also, it might be a god solution to have a list of abbreviations.

Reply: BC and RC are changed to Brayton and Rankine cycle. Only CCGT and PCGT are used which are easily comprehensible.

Best regards

Reviewer 3 Report

This is an interesting article that worth's publication.

My only remark is that the authors should provide better quality figures

(they do not appear very well at least in the pdf)

Author Response

Dear Reviewer

Thank you for reviewing the article

Matlab graphs are of poor quality. I replaced their axis.

regards

Reviewer 4 Report

The paper is presented to a high quality, with a good standard of English.

The concept of a PCGT is interesting and the paper demonstrates the possible advantages for different working fluids.

The weakness of the approach adopted, within the tool set deployed, is the use of a simplified model of the new combustion chamber/heat exchanger. As the authors comment, CFD would be the a more appropriate methodology to employ.

As such, the conclusions drawn by the paper can only be made under the proviso that a more detailed design analysis of the combustion chamber/heat exchanger would be undertaken. The principle issues that must be considered include the maximum working temperature of available high temperature materials (~1300K, for uncooled nickel alloys), overall combustion efficiency and the minimisation of NOx and soot production through control of air-fuel ratio and the potential use of partial/full premixing. This itself then further introduces possible issues of combustion instability.

Author Response

Dear Reviewer

Thank you for reviewing the article

I agree 100% that the exact analysis/design should be done using CFD calculations. The scope of this article is to only evaluate the opportunity thermodynamically. It provides the thermodynamic limits/boundaries that should be considered while using CFD.

In future, we plan to collaborate with some other group to help us in the design of the combustor.

Best regards

Round 2

Reviewer 1 Report

Accept 

Author Response

Dear Reviewer

I checked the manuscript and corrected some spellings.

Best regards

Reviewer 2 Report

Report of revised version of „Thermodynamic Analysis of Partitioned Combined Cycle using Simple Gases” by Aqeel Ahmad Taimoor, Ehtisham Siddiqui , Salem S. Abdel Aziz

The revised version of the manuscript is improved and now it is much better. The problem about the use of one or another equation of states are properly addressed in the new version and most of the other problems are also solved.

The quality of figures are also improved, labels and numbers are bigger, but the graphs are still very blurred. Even reading the manuscript on the monitor with some magnification, some of the figures are not clear. In this way, Readers might not understand important parts. I would recommend some more improvement of the figures. I know that the normal figures for Mathlab are low-resolution ones, but there are several ways to improve the quality and resolution.

Decision: minor revision

Author Response

Dear Reviewer

I must admit that I am not very good at editing figures for high-quality resolution. However, with the help of one of my colleague, the images are sharpened compared to previous ones.

Some minor spelling mistakes are corrected.

Best regards